# An Incidental Finding of Appendiceal Gastrointestinal Stromal Tumor with Abundant Skeinoid Fibers: A Rare Case Report with Insights from a Comprehensive Literature Review

**DOI:** 10.3390/diagnostics15070924

**Published:** 2025-04-03

**Authors:** Yu Liu, Yaomin Chen, Asra Feroze, Zhiyan Fu

**Affiliations:** Department of Pathology, LSUHSC School of Medicine, New Orleans, LA 70112, USA; yliu6@lsuhsc.edu (Y.L.); ychen3@lsuhsc.edu (Y.C.); aferoz@lsuhsc.edu (A.F.)

**Keywords:** appendix, gastrointestinal stromal tumors (GISTs), skeinoid fibers, CD117, DOG1

## Abstract

**Background and Clinical Significance:** Gastrointestinal stromal tumors (GISTs) are the most common mesenchymal tumors of the gastrointestinal tract but are exceedingly rare in the appendix, with only 20 cases reported in the literature. Due to their rarity, clinical behavior, histopathologic features, and management of appendiceal GISTs remain poorly understood. **Case Presentation**: We present the case of a 74-year-old man who underwent a right hemicolectomy for colonic adenocarcinoma, during which an incidental 1.2 cm appendiceal GIST was discovered. Histopathological examination revealed spindle cell morphology with abundant skeinoid fibers (SF), minimal mitotic activity (<1/50 HPF), and no nuclear atypia. Immunohistochemical staining confirmed positivity for CD117, DOG1, and CD34. The tumor was classified as low risk based on its size and mitotic count, and the patient remained recurrence-free at a 4-month follow-up. **Conclusions:** Our case expands the limited literature on appendiceal GISTs by demonstrating their histopathological and immunohistochemical features, favorable prognostic outcomes, and potential for incidental detection during surgeries for unrelated conditions. However, additional studies are needed to further elucidate their molecular characteristics and overall clinical behavior.

## 1. Introduction

Gastrointestinal stromal tumors (GISTs) are the most common mesenchymal neoplasms of the gastrointestinal tract, typically arising in the stomach, small intestine, or rectum, with fewer cases reported in the esophagus, colon, omentum, and mesentery [1]. They originate from the interstitial cells of Cajal and are primarily driven by activating mutations in the *KIT* or *PDGFRA* genes, leading to uncontrolled cell proliferation via constitutive activation of tyrosine kinase signaling pathways. The introduction of imatinib mesylate, a tyrosine kinase inhibitor targeting both KIT and PDGFRA receptors, has significantly improved outcomes in patients with advanced GISTs [2]. In contrast, GISTs originating from the appendix are exceedingly rare, with only 20 cases reported in the literature to date. Due to their rarity, their clinical behavior, histopathologic features, and management remain poorly understood. In this report, we describe a rare case of an incidental appendiceal GIST with abundant skeinoid fibers, discovered during surgery for colonic adenocarcinoma. We also provide a comprehensive review of the literature to further elucidate the clinicopathologic, molecular, and prognostic features of this uncommon entity.

## 2. Case Presentation

A 74-year-old African American man with a history of diabetes mellitus and hypertension underwent a routine screening colonoscopy, which revealed a colonic mass near the hepatic flexure. Biopsy findings confirmed at least high-grade dysplasia/intramucosal adenocarcinoma. An abdominal CT scan demonstrated a 3.7 cm mass in the hepatic flexure of the colon, with a normal appendix.

The patient subsequently underwent a right hemicolectomy. Gross examination revealed a red–tan, firm, polypoid mass measuring 3.7 × 1.8 × 1.6 cm in the colon at the hepatic flexure, with invasion into the muscularis propria and involvement of the pericolonic adipose tissue. The histopathological evaluation confirmed the diagnosis of moderately differentiated invasive adenocarcinoma (pT3N2a), with four out of twenty-nine lymph nodes positive for metastatic carcinoma.

Incidentally, serial sectioning of the appendix revealed a firm, white–tan area surrounding the pinpoint lumen in the proximal portion, measuring 1.2 × 1.1 × 0.9 cm. Histopathological examination of the appendiceal lesion identified spindle cells (Figure 1) with PAS-positive extracellular collagen globules (Figure 2, consistent with skeinoid fibers. No nuclear atypia or necrosis was observed, and mitotic activity was less than 1 per 50 high-power fields. Immunohistochemical analysis (Figure 3) demonstrated that the lesional cells were positive for CD117, DOG1, CD34, and vimentin, while negative for SMA, Desmin, and S100. These findings supported the diagnosis of a gastrointestinal stromal tumor (GIST) of the appendix, of spindle cell type with skeinoid fibers. There were no microscopic signs of appendicitis, and all margins of the appendix were negative for tumor involvement.

The patient remained alive and well at the 4-month follow-up.

## 3. Discussion and Literature Review

Appendiceal GISTs are extremely rare, accounting for only 0.1% of all GISTs in the dataset [1]. To date, only 20 cases originating from the appendix have been described in the literature (Table 1). The majority of appendiceal GISTs occur in older adults, with a mean age of 63.7 years (ranging from 6 to 86 years), including our case, although a single pediatric case has been reported in a 6-year-old boy [3]. There is a marked male predominance, with a male-to-female ratio of approximately 3:1. Our case contributes valuable evidence to the limited understanding of appendiceal GISTs and provides additional insight into this rare clinical entity.

Although 55% of patients initially present with symptoms mimicking acute appendicitis, in most reported cases the histopathologic examination does not confirm acute inflammation. Instead, 30% of appendiceal GISTs are instead detected incidentally during surgeries for unrelated pathologies. Approximately 30% of patients with appendiceal GIST have concurrent malignancies, such as bladder carcinoma, cervical cancer, pulmonary adenocarcinoma, endometrial carcinoma, or mantle cell lymphoma [4,5,6,7,8]. In our case, the appendiceal GIST coexisted with colonic adenocarcinoma and was discovered incidentally during grossing examination after surgery. However, due to the limited number of reported cases, the clinical significance of these associations remains unclear and warrants further investigation. One unique case [6] involved simultaneous appendiceal and gastric GISTs. The appendiceal GIST exhibited a spindle cell morphology with involvement of the muscular wall, whereas the gastric GIST displayed an epithelioid phenotype and subsequently metastasized to the liver. These distinct histological differences strongly suggested that the two GISTs were entirely independent tumors. Another case [8] described an appendiceal GIST coexisting with neurofibromatosis type 2 (NF2). In contrast, neurofibromatosis type 1 (NF1) is more frequently associated with GIST; however, appendiceal involvement has not been discovered. A study by Miettinen and Fetsch et al. [9], which analyzed 45 patients with NF1-associated GISTs, found that the majority of tumors arose in the jejunum, ileum, or duodenum, with 28 patients presenting with multiple GISTs. None of these cases involved the appendix. No clear relationship has been established between NF2 and appendiceal GIST. Given that only one instance of appendiceal GIST coexisting with NF2 has been identified among the 20 reported cases, this co-occurrence is likely coincidental.

With respect to anatomical distribution, appendiceal GISTs can arise anywhere along the appendix, including the tip (5/20), the proximal portion (6/20), the mid-portion (4/20), or the mesoappendix (1/20). Their sizes range from very small lesions (<1 cm) to tumors exceeding 10 cm in their greatest dimension. Histologically, most appendiceal GISTs exhibit a spindle cell pattern, with occasional epithelioid components (2/20). Of the twenty reported cases, only five specifically mention the presence of skeinoid fibers [4,6,10]. Our case also demonstrates a substantial presence of skeinoid fibers. All six appendiceal GISTs with skeinoid fibers (6/21), including our case, were classified as low risk or very low risk according to modified NIH risk classification [11]. The average patient age was 65.3 years (range: 56–78 years), with a male predominance (M:F ratio of 5:1). Four cases were located in the proximal portion of the appendix, one at the tip, and one case had an unspecified location. A comprehensive study by Miettinen and Makhlouf et al. [12], which analyzed 906 small intestinal GISTs, identified skeinoid fibers in 44% of cases. Their presence was linked to a more favorable clinical course, suggesting that these fibers might serve as a beneficial prognostic marker. However, further research is needed to confirm this potential correlation and clarify its clinical significance in appendiceal GISTs.

Consistent with classic GIST phenotypes, nearly all appendiceal GISTs are immunoreactive for CD117 (c-KIT) and/or DOG1, with frequent co-expression of CD34. Muscle and neural markers (SMA, Desmin, S-100) are typically negative, although focal S-100 positivity has been noted in two cases [7,8]. This distinct immunohistochemical profile allows for confident differentiation of GISTs from more common appendiceal tumors, such as neuroendocrine tumors, leiomyomas, schwannomas, and mucinous epithelial neoplasms. Among the 20 reported cases of appendiceal GISTs, mutation analysis has been performed in only five [4,5,7,13]. In the present case, gene mutation testing was not performed. Of the five analyzed cases, three tumors revealed heterozygous *KIT* exon 11 mutations in the juxtamembrane region: Case 1 exhibited a missense mutation (K558R), Case 2 harbored an in-frame deletion (I571_R588), and Case 3 had a deletion/inversion mutation (K588_V559). The remaining two tumors were wild-type for *KIT*. However, non-appendiceal gastrointestinal GISTs are typically sporadic and exhibit a broader mutation spectrum, including alterations in *KIT* (exons 2, 9, 11, 13, 14, and 17) and *PDGFRA* (exons 12 and 14), as well as less common mutations involving *SDH*, *NF1*, *BRAF*, and *KRAS* [2,5,14,15]. In contrast, pediatric GISTs arising outside the appendix more frequently occur in syndromic or familial contexts, such as Carney–Stratakis syndrome or Carney’s triad, and are often associated with deficiencies in the succinate dehydrogenase (SDH) complex. *KIT* or *PDGFRA* mutations are identified in only about 15% of pediatric and young adult cases [16,17]. However, no familial syndrome or genetic mutation has been identified in the single reported pediatric case of appendiceal GIST. This likely reflects the limited number of appendiceal GISTs both in adults and children that have been studied at the molecular level.

Risk stratification of appendiceal GISTs generally follows the same principles applied to small bowel GISTs, where size and mitotic rate are the primary determinants of malignant potential. Smaller tumors (<2 cm) with low mitotic activity (<5 mitoses per 50 HPF) typically fall into the low- or very-low-risk categories and generally respond well to local resection alone. In contrast, larger tumors (≥5 cm) and/or those with high mitotic counts tend to exhibit more aggressive behavior. Among the twenty reported cases, three presented with malignant features: one involving a six-year-old patient with increased mitosis (12/50 HPF) [3], another with peritoneal metastasis [18], and a third case of appendiceal GIST invading locally and perforating adjacent bowel [19]. In these cases, neoadjuvant or adjuvant therapy with tyrosine kinase inhibitors such as imatinib effectively reduced tumor burden and prevented recurrence.

**Table 1 diagnostics-15-00924-t001:** Clinicopathologic and molecular features of reported appendiceal GISTs.

Case No.	Reference	Age	Gender	Symptoms	Location	Size (cm)	Histology	Risk Classification	Coexisting Malignancy	Genotype	Outcome
1	Markku Miettinen et al., 2001 [6]	64	M	Incidental (autopsy)	Tip	1.4	Spindle cell: SF+	Low risk	Metastatic pulmonary adenocarcinoma	N/A	Died (unrelated)
2	Markku Miettinen et al., 2001 [6]	56	M	Appendicitis-like	Proximal	1.2	Spindle cell; SF+	Low risk	None	N/A	Alive at 8 Y
3	Markku Miettinen et al., 2001 [6]	59	M	Incidental	N/A	0.9	Spindle cell; SF+	Low risk	Gastric epithelioid GIST	N/A	Died at 15 Y (liver metastasis from gastric GIST)
4	Markku Miettinen et al., 2001 [6]	72	M	Appendicitis	Proximal	1.3	Spindle cell	N/A	None	N/A	Died at 4 Y (COPD)
5	W. M. Yap et al., 2005 [8]	66	F	Appendicitis-like	Mid portion	0.25	Spindle cell	Low risk	NF2	N/A	N/A
6	Kyu-Jong Kim et al., 2007 [20]	56	M	GI bleeding	Mid portion	<1	Spindle cell	Low risk	None	N/A	No recurrence at 3 M
7	A. Agaimy et al., 2008 [4]	72	M	Incidental	Tip	2.5	Spindle cell; focal epithelioid	Low risk	Urinary bladder carcinoma	KIT inframe deletion I571_R588 (Exon 11)	No recurrence at 4 M
8	A. Agaimy et al., 2008 [4]	78	F	Appendicitis-like	Proximal	0.5	Spindle cell; SF+	Very low risk	Endometrial adenocarcinoma	KIT K558R (Exon11)	N/A
9	A. Agaimy et al., 2008 [5]	86	F	N/A	N/A	<1	Spindle cell	Very low risk	None	Wild type	N/A
10	Kurosh Rahimi et al., 2009 [7]	65	F	Incidental	N/A	1.1	Epithelioid and spindle cell; nerve differentiation	Very low risk	Mantle cell lymphoma with appendiceal involvement	Wild type	No recurrence at 24 M
11	Ram Elazary et al. [19]	57	M	Abdominal pain and peri-appendiceal abscess	Tip	20	Spindle cells	Malignant (invading adjacent bowel)	None	N/A	N/A
12	Wenhua Li et al., 2012 [3]	6	M	Abdominal pain and hard mass	Mesoappendix	6.7	Spindle and epithelioid cell	Malignant	None	N/A	No recurrence at 9 M
13	M. Bouassida et al., 2013 [21]	75	M	Appendicitis, perforation, and acute peritonitis	Mid portion	2.0	Spindle cell	Low risk	None	N/A	No recurrence at 48 M
14	Nikolaos Vassos et al., 2013 [13]	48	M	Appendicitis-like	N/A	3.0	Spindle cell	High risk (tumor rupture)	None	KIT 558-559del and 559-560ins (Exon11)	No recurrence at 27 M
15	J. M. Chun and K. H. Lim, 2016 [22]	68	M	Appendicitis	Proximal	3.0	Spindle cell	Low risk	None	N/A	No recurrence at 35 M
16	M. Kaneko et al., 2017 [18]	67	M	Large abdominal mass	Tip	22.0	Spindle cell	Malignant (peritoneal metastasis)	None	N/A	No recurrence at 6 M
17	Bao Zhang et al., 2018 [23]	59	F	Incidental	N/A	10	N/A	High risk	Cervical cancer	N/A	N/A
18	N. Mourra et al., 2020 [10]	61	M	Appendicitis-like symptoms	Proximal	1.0	Spindle cell; SF+	Very low risk	None	N/A	N/A
19	Ahmad Elnassasra et al., 2022 [24]	75	M	Appendicitis	Tip	<1	Spindle cell	Low risk	None	N/A	N/A
20	Jacob D. Williams et al., 2024 [25]	74	M	Abdominal pain; appendiceal mass	Mid portion	1.5	Spindle cell	Low risk	None	N/A	No recurrence at 2 M
	Our case	74	M	Incidental	Proximal	1.2	Spindle cell; SF+	Very low risk	Colonic adenocarcinoma	N/A	No recurrence at 4 M

## 4. Conclusions

Appendiceal GISTs are exceptionally rare gastrointestinal stromal tumors, with 30% of cases being detected incidentally. Skeinoid fibers have been identified in only six cases to date and may serve as a favorable prognostic marker for these tumors, similar to small bowel GISTs. Although molecular studies have identified *KIT* exon 11 mutations in appendiceal GISTs, further research is needed to elucidate their broader genetic landscape and biological behavior.

## Figures and Tables

**Figure 1 diagnostics-15-00924-f001:**
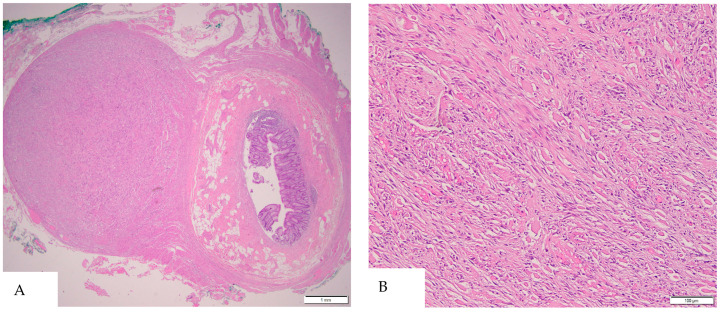
**(A**) Low-power view reveals a cellular nodule within the muscularis propria layer of the appendix (H&E, 20×). (**B**) High-power view shows the appendiceal GIST as a spindle cell tumor infiltrating between the smooth muscle fibers, without nuclear atypia or mitosis (H&E, 200×).

**Figure 2 diagnostics-15-00924-f002:**
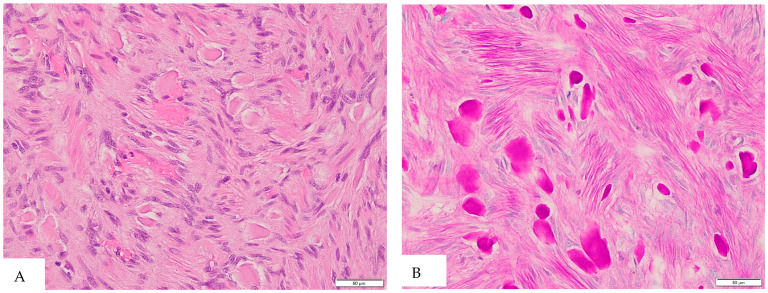
(**A**) H&E staining at 400× magnification reveals abundant extracellular collagen globules (skeinoid fibers) in the appendiceal GIST. (**B**) Periodic acid–Schiff (PAS) staining demonstrates scattered PAS-positive collagen globules, consistent with skeinoid fibers.

**Figure 3 diagnostics-15-00924-f003:**
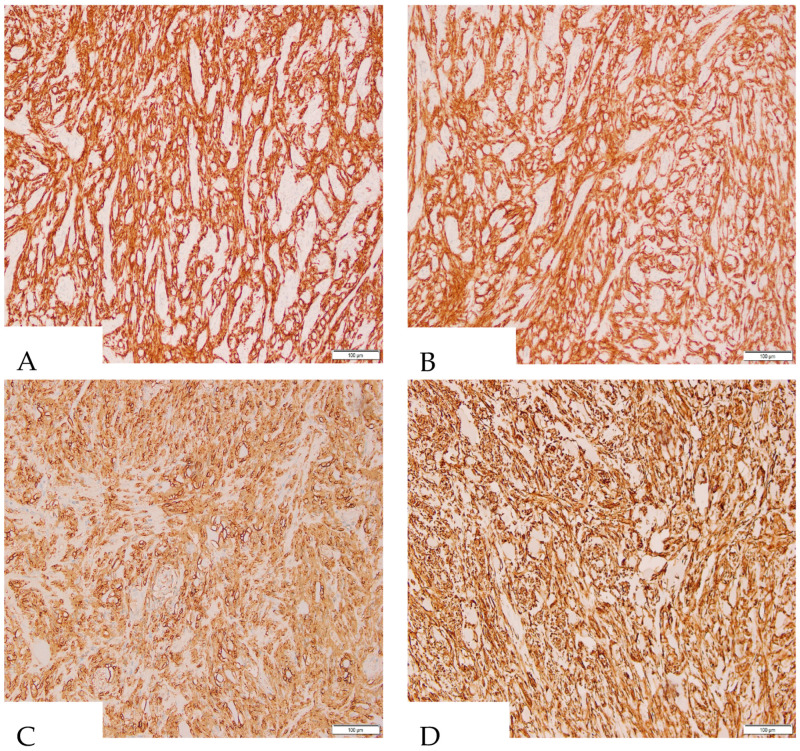
Immunohistochemical staining at 200× magnification demonstrates strong positivity of tumor cells for DOG1 (**A**), CD117 (**B**), CD34 (**C**), and Vimentin (**D**).

## Data Availability

The original contributions presented in this study are included in the article. Further inquiries can be directed to the corresponding author.

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
