# Peer review of "An Incidental Finding of Appendiceal Gastrointestinal Stromal Tumor with Abundant Skeinoid Fibers: A Rare Case Report with Insights from a Comprehensive Literature Review"

_diagnostics, 2025, doi:10.3390/diagnostics15070924_

Round 1

Reviewer 1 Report

Comments and Suggestions for Authors

Minor changes are recommended: You should go straight to the point, use fewer pictures, and write a more synthetic discussion and analysis of key features of the cases. I would also suggest a shorter bibliography, as it is too long for a case report.

Author Response

Comment 1: Minor changes are recommended: You should go straight to the point, use fewer pictures, and write a more synthetic discussion and analysis of key features of the cases. I would also suggest a shorter bibliography, as it is too long for a case report.

Response 1:  Thank you very much for your constructive feedback. In response to your suggestion, we have shortened the summary table by streamlining the content and focusing on key clinicopathologic features (please see the attachment). While we have retained a detailed discussion and analysis of the reported cases, we believe the revised table format enhances clarity and readability. Additionally, the bibliography includes all previously published case reports on appendiceal GISTs—over 20 in total—as we performed a comprehensive literature review to capture every relevant case. Given the extreme rarity of appendiceal GISTs and the limited number of published cases, we respectfully chose to maintain the current level of detail in the manuscript to provide a comprehensive and clinically meaningful contribution to the understanding of this exceptionally rare entity. We sincerely appreciate your thoughtful input and kind understanding.

Reviewer 2 Report

Comments and Suggestions for Authors

This case describes a GIST of appendix, found during the resection of right colon because of adenocarcinoma. The text is well written, it is easy to read and understand.

Comments:

(1) Is it possible to show the colonoscopy mass that was found in the hepatic flexure?

(2) Do you have access to histological image of the adenocarcinoma (line 41)?

(3) Line 52. Why are the pas-positive extracellular collagen globules/skenoid fibers important?

(4) Line 58. Please define microscopic features of appendicitis.

(5) Line 77. Regarding " These co-occurrences may reflect underlying shared risk factors or genetic predispositions". Please elaborate.

(6) As I understand. This case has not mutational analysis of KIT.

(7) You may expand the differential diagnosis with other neoplasia.

(8) You may mention difference between GIST of adults and pediatric patients (genetic predisposition, genetic syndromes).

(9) You may expand the description of the pathogenesis of GIST. For example, the postulated cell of origin is likely the cells of Cajal (ICCs).

(10) Regarding the genetic syndromes, it may be worth mentioning the primary familial gist syndrome, NF1, pediatric and AYA patients, Carney-stratakis syndrome, carney triad, etc.

(11) How often is the clinical presentation as metastatic disease?

(12) You may expand with the description of paraneoplastic syndromes in adult patients with GIST

(13) You may describe/discuss the PDGFRA mutations, and other like BRAF, NF-1, NTRK, etc.

Author Response

Comment 1: Is it possible to show the colonoscopy mass that was found in the hepatic flexure?

Response 1: Thank you very much for your thoughtful suggestion. Unfortunately, colonoscopy images of the hepatic flexure mass are not available for this case. We regret that we cannot include these images in the revised manuscript.

Comment 2: Do you have access to histological images of the adenocarcinoma (line 41)?

Response 2: Thank you very much for your valuable suggestion. We do have histological images of the adenocarcinoma. Initially, we did not include these images as our manuscript primarily focuses on appendiceal GIST. However, if you believe these images would enhance the manuscript, we are happy to add them. Please let us know your preference. Thank you again for your helpful advice.

Comment 3: Line 52. Why are the PAS-positive extracellular collagen globules/skeinoid fibers important?

Response 3: We greatly appreciate your insightful comment. Skeinoid fibers are thought to hold potential prognostic significance. A comprehensive study analyzing 906 small intestinal GISTs identified skeinoid fibers in 44% of cases, correlating their presence with a more favorable clinical outcome. However, appendiceal GISTs with skeinoid fibers are rare, with only six reported cases including ours. Therefore, further research is needed to confirm their prognostic relevance specifically in appendiceal GIST. We have clarified this point in the discussion section of the revised manuscript (discussion: paragraph 3, lines 82-90) and highlighted it in red.

Comment 4: Line 58. Please define microscopic features of appendicitis.

Response 4: Thank you very much for your suggestion. In our case, no microscopic features indicative of appendicitis were observed. We have clearly stated this absence in the revised manuscript and highlighted the relevant text in red (case presentation section, paragraph 3, line 50).

Comment 5: Line 77. Regarding "These co-occurrences may reflect underlying shared risk factors or genetic predispositions," please elaborate.

Response 5: Thank you very much for your insightful comment. Given the limited literature, the association between appendiceal GISTs and concurrent malignancies remains unclear. However, it is plausible that the presence of an additional malignancy may indicate an underlying predisposition to tumorigenesis. This could potentially be attributed to shared genetic alterations, common environmental exposures, or systemic factors that influence overall cancer susceptibility. However, this hypothesis requires further investigation. We have clarified this point in the revised manuscript (Discussion, paragraph 2, lines 65-68), and the changes have been highlighted in red.

Comment 6: As I understand, this case has no mutational analysis of KIT.

Response 6: Thank you very much for pointing this out. We confirm that KIT mutational analysis was not performed in this case. This has been noted in the revised manuscript, highlighted in red (discussion, paragraph 4, line 96)

Comment 7: You may expand the differential diagnosis with other neoplasia.

Response 7: Thank you for your valuable comment. In response, we have briefly mentioned relevant differential diagnoses in the revised discussion section (paragraph 4, lines 93-95), emphasizing that appendiceal GISTs can be confidently distinguished from other neoplasms—such as neuroendocrine tumors, leiomyomas, schwannomas, and mucinous epithelial neoplasms —based on their distinct immunohistochemical profile. This addition is clearly highlighted in red.

Comment 8: You may mention the differences between GIST in adults and pediatric patients (genetic predisposition, genetic syndromes).

Response 8: Thank you very much for your constructive recommendation. We fully agree and have added a discussion highlighting the differences of the genetic predispositions and associated genetic syndromes between adult and pediatric GISTs. This addition has been clearly highlighted in red in the revised manuscript (discussion, paragraph 4, lines 99-108).

Comment 9: You may expand the description of the pathogenesis of GIST. For example, the postulated cell of origin is likely the cells of Cajal (ICCs).

Response 9: Thank you very much for your valuable guidance. We fully agree and have expanded our description of GIST pathogenesis in the introduction section (paragraph 1, lines 24-27) and highlighted it in red.

Comment 10: Regarding genetic syndromes, it may be worth mentioning the primary familial GIST syndrome, NF1, pediatric and AYA patients, Carney-Stratakis syndrome, Carney triad, etc.

Response 10: Thank you very much for your valuable comment. The genetic syndromes associated with GIST—including primary familial GIST syndrome, Carney-Stratakis syndrome, and Carney triad—have already been discussed in the section comparing adult and pediatric GISTs. This content has been included in the revised manuscript and clearly highlighted in red (discussion section, paragraph 4, lines 99-108).

Comment 11: How often is the clinical presentation metastatic disease?

Response 11: Thank you very much for highlighting this important point. Among 21 reported cases of appendiceal GIST, only one displays peritoneal metastasis. We have included this information clearly in the revised manuscript (discussion, paragraph 5, lines 114), highlighted in red.

Comment 12: You may expand with the description of paraneoplastic syndromes in adult patients with GIST.

Response 12: Thank you very much for your helpful suggestion. Paraneoplastic syndromes are exceedingly rare in association with GISTs, and among the 20 previously reported cases of appendiceal GIST, none have described any accompanying paraneoplastic manifestations. Therefore, we did not include this aspect in our discussion. Nonetheless, we sincerely appreciate your valuable comment.

Comment 13: You may describe/discuss PDGFRA mutations, and others like BRAF, NF-1, NTRK, etc.

Response 13: Thank you very much for your insightful suggestion. We agree that mutations such as PDGFRA, BRAF, NF1, and NTRK are important mutations of the molecular landscape of GISTs, despite their relatively rare occurrence. In our manuscript, we have briefly discussed these genetic alterations in the context of non-appendiceal GISTs. However, according to the current literature, appendiceal GISTs have only been reported with KIT exon 11 mutations, and no other mutations, including PDGFRA or BRAF, have been identified in this rare subset. Therefore, to maintain a focused and concise discussion relevant to appendiceal GISTs, we have not expanded this section further. We sincerely appreciate your valuable comments and understanding.
